# Predictive Value of Early Post-Treatment Diffusion-Weighted MRI for Recurrence or Tumor Progression of Head and Neck Squamous Cell Carcinoma Treated with Chemo-Radiotherapy

**DOI:** 10.3390/cancers12051234

**Published:** 2020-05-14

**Authors:** Esteban Brenet, Coralie Barbe, Christine Hoeffel, Xavier Dubernard, Jean-Claude Merol, Léa Fath, Stéphanie Servagi-Vernat, Marc Labrousse

**Affiliations:** 1Department of Oto-Rhino-Laryngology, Head and Neck Surgery, Robert Debré University Hospital, 51100 Reims, France; xdubernard@chu-reims.fr (X.D.); jcmerol@chu-reims.fr (J.-C.M.); mlabrousse@chu-reims.fr (M.L.); 2Clinical Research Unit, Robert Debré University Hospital, 51100 Reims, France; cbarbe@chu-reims.fr; 3Department of Radiology, Robert Debré University Hospital, 51100 Reims, France; choeffel-fornes@chu-reims.fr; 4Department of Oto-Rhino-Laryngology, Head and Neck Surgery, University Hospital of Strasbourg, 67000 Strasbourg, France; lea.fath@chru-strasbourg.fr; 5Department of Radiotherapy, Institut Godinot, 51100 Reims, France; stephanie.servagivernat@reims.unicancer.fr

**Keywords:** head and neck neoplasm, diffusion-weighted magnetic resonance imaging (MRI), recurrence, chemo-radiotherapy

## Abstract

Aims: To investigate the predictive capacity of early post-treatment diffusion-weighted magnetic resonance imaging (MRI) for recurrence or tumor progression in patients with no tumor residue after chemo-radiotherapy (CRT) for head and neck squamous cell carcinoma, and, to assess the predictive capacity of pre-treatment diffusion-weighted MRI for persistent tumor residue post-CRT. Materials and Method: A single center cohort study was performed in one French hospital. All patients with squamous cell carcinoma receiving CRT (no surgical indication) were included. Two diffusion-weighted MRI were performed: one within 8 days before CRT and one 3 months after completing CRT with determination of median tumor apparent diffusion coefficient (ADC). Main outcome: The primary endpoint was progression-free survival. Results: 59 patients were included prior to CRT and 46 (78.0%) completed CRT. A post-CRT tumor residue was found in 19/46 (41.3%) patients. In univariate analysis, initial ADC was significantly lower in patients with residue post CRT (0.56 ± 0.11 versus 0.79 ± 0.13; *p* < 0.001). When initial ADC was dichotomized at the median, initial ADC lower than 0.7 was significantly more frequent in patients with residue post CRT (73.7% versus 11.1%, *p* < 0.0001). In multivariate analysis, only initial ADC lower than 0.7 was significantly associated with tumor residue (OR = 22.6; IC [4.9–103.6], *p* < 0.0001). Among 26 patients without tumor residue after CRT and followed up until 12 months, 6 (23.1%) presented recurrence or progression. Only univariate analysis was performed due to a small number of events. The only factor significantly associated with disease progression or early recurrence was the delta ADC (*p* = 0.0009). When ADC variation was dichotomized at the median, patients with ADC variation greater than 0.7 had time of disease-free survival significantly longer than patients with ADC variation lower than 0.7 (377.5 [286–402] days versus 253 [198–370], *p* < 0.0001). Conclusion and relevance: Diffusion-weighted MRI could be a technique that enables differentiation of patients with high potential for early recurrence for whom intensive post-CRT monitoring is mandatory. Prospective studies with more inclusions would be necessary to validate our results.

## 1. Introduction

Head and neck cancer is the fifth most common cancer, representing 5.3% of all cancers, with 890,000 new cases worldwide in 2017 [1,2]. 

Prognosis is frequently poor and is closely correlated with tumor status. Most cases are diagnosed at a locally advanced stage [3]. Chemo-radiotherapy (CRT) protocols are standard first line treatment for head and neck squamous cell carcinoma (HNSCC), either as part of an organ preservation strategy or in cases where surgery has initially been ruled out, as is the case for locally advanced tumors [4,5,6,7,8,9,10,11,12,13,14].

Furthermore, these cancers characteristically have a high recurrence rate, approximately 25% [15]. Diagnosis of local and lymph node recurrence is a vital part of monitoring patients under treatment for HNSCC. Such monitoring should be particularly intensive for patients who are most at risk, but more importantly, for patients who are candidates for curative treatment [16,17]. The purpose of early diagnosis of locoregional recurrence is to offer curative treatment [18]. This is because local recurrence promotes lymph node recurrence which in turn promotes onset of metastasis [19,20]. Very early diagnosis of recurrence facilitates curative treatment and improvement in quality of life [19,21,22]. 

Yet, within the context of follow-up care in these patients, monitoring procedures are at times suggestive of suspected tumor recurrence without diagnostic support from clinical examination or conventional imaging assessment. Treatment-related mucosal changes occasionally complicate identification of this type of recurrence, from both a clinical and a radiological standpoint. In which case, a wait-and-see attitude should be adopted. 

Diffusion-weighted imaging (DWI) differentiates tissues on the basis of molecular water mobility on a microscopic scale, which is indirectly determined by tissue cellularity. Apparent diffusion coefficient (ADC) measures the rate of diffusion of water molecules within a tissue. A low value for DWI/ADC indicates that the tissue is well organized, while a high value for ADC indicates that the tissue is not well organized [23].

Numerous reports have shown the usefulness of DWI/ADC in oncology [24,25,26,27]. According to the literature, ADC values can discriminate malignant and benign lesions [28,29,30]. Usually, malignant tumors have lower values in comparison to benign lesions [31,32,33]. Several studies have suggested that the characteristics of tumors on magnetic resonance imaging (MRI)/DWI, especially ADC values, commonly correlate with histologic characteristics known as poor prognosis in HNSCC as well as in rectal, pelvic, cerebral and breast tumors [29,32,34,35,36], and thus predict both long-term survival and resistance to CRT [37]. Other studies have suggested that MRI/DWI is a potentially valuable tool for early detection of responders to chemotherapy, thus enabling rapid adjustment to the therapeutic strategy of non-responders during treatment [38,39]. Changes in the diffusion coefficient occur sooner than morphological changes visible on conventional imaging, generating a significantly higher rise in ADC values in responders than in non-responders [26,40].

The importance of MRI/DWI in predicting early recurrence or disease progression soon after completion of CRT has yet to be assessed. 

The main aim of the present study was to investigate the predictive capacity of early post-treatment high b-value images derived from DWI, with particular focus on ADC variation between pre-treatment diffusion-weighted MRI and post-treatment diffusion-weighted MRI (delta ADC), for recurrence or tumor progression in patients with no tumor residue 3 months post treatment by CRT for primary HNSCC. 

The secondary aim was to assess the predictive capacity of pre-treatment high b-value images derived from DWI in the absence of total response to CRT. 

## 2. Materials and Methods

### 2.1. Patients

A single center cohort study was performed between April 2014 and April 2018. All patients older than 18 years old, beginning CRT for not operable or not resectable HNSCC, or with laryngeal/hypopharyngeal cancer, for which organ preservation was decided in Reims university hospital, and who accepted to participate in the study, were included. Patients with contra-indication to MRI were excluded.

Every patient that agreed to participate in the study provided informed written consent. This study was approved by the Ethics Committee (CPP Est III, Nancy, 5 November 2013) and retrospectively registered in clinicaltrials.gov (NCT02862678; 11 August 2016).

### 2.2. Data Collection and Outcomes

Baseline patients’ characteristics (age, sex, tobacco, alcohol consumption) and baseline tumor characteristics (size, location, Tumor/Node/Metastasis classification (TNM) stage) were recorded. 

Concerning patients without tumor residue after completing CRT, a 12-month follow-up was performed. Tumors were considered as recurrent if, from 6 months post-treatment onwards, histology specimens prompted by clinical and radiological findings confirmed squamous cell carcinoma. Tumors were considered as progressive if, between 3 and 6 months post-treatment, histology specimens prompted by clinical and radiological findings confirmed squamous cell carcinoma. 

### 2.3. Diffusion-Weighted MRI

Two high b-value images derived from DWI scans were performed for each patient: one prior to initiating CRT, a maximum of 8 days before starting the CRT protocol, including determination of tumor ADC (ADC1), and one within 3 months of CRT completion, including determination of tumor ADC (ADC2).

MRI/DWI sequences were all obtained from the same university teaching hospital MRI scanner: the 3 Tesla MRI system (MAGNETOM, Avanto, Siemens^®^). Routine protocol was followed, combining T1-weighted, T2-weighted and T1-weighted with gadolinium contrast injection sequences (standard MRI). Diffusion-weighted sequences were then acquired with a voxel size of 1.9 × 1.9 × 3.0 mm, 3-gradient diffusion weighting, and b1 value = 0 s/mm^2^, b2 value = 500 s/mm^2^ and b3 value = 1000 s/mm^2^ (Figure 1).

The sequences thus obtained were then analyzed using COME VE10F, NUMARIS/4, Siemens^®^ software (Siemens AG, Muenchen, Germany).

The DWI/ADC coefficient was calculated by two different radiologists with experience of more than 5 years in imaging the head and neck. Readers were blinded from each over and from clinical results. 

The measurement of DWI/ADC was carried out on the whole of the tumor, including on the areas of necrosis on the surface of the tumor on 3 sections with the largest diameter in axial sections. 3 measurements were performed for each tumor and the ADC mean coefficient was chosen [41,42].

### 2.4. Statistical Analysis

Data were described as mean and standard deviation (SD) for quantitative variables and number and percentage for qualitative variables. 

The inter-reader reproducibility for DWI/ADC measurement was studied using intra-class correlation coefficient (ICC). An ICC > 0.7 indicates good reproducibility [43].

Factors associated with recurrence or disease progression, notably delta ADC, were studied using univariate analysis (Log-rank tests). No multivariate analysis could be conducted due to the small sample size of the study. Median delta ADC of all patients without tumor residue after completing CRT was used as a threshold to optimally differentiate patients with and patients without recurrence or disease progression. 

Factors associated with presence of tumor residue, notably ADC1 value of the tumor, were studied using univariate analysis (Wilcoxon, Chi-square and Fisher’s exact tests, as appropriate) and multivariate analysis (stepwise logistic regression with entry and removal thresholds of 0.20 and variables with a *p* < 0.20 by univariate analysis included). Median ADC1 of all patients completing CRT was used as a threshold to optimally differentiate patients with and patients without post-CRT tumor residue. 

Sensitivities, specificities, positive predictive values (PPV) and negative predictive values (NPV) with 95% confidence intervals (CIs) were calculated for each ADC threshold.

A *p* value < 0.05 was considered statistically significant. All analyses were performed using SAS version 9.4 (SAS Inc., Cary, NC, USA).

## 3. Results

### 3.1. Patients’ Characteristics, Treatment and Follow-up

The flow-chart of the study is shown in Figure 2. Between 4 April 2014 and 22 April 2018, 59 patients with HNSCC were included. The first diffusion-weighted MRI scan was performed within the 8 days prior to treatment initiation for all patients. Thirteen patients (22%) died during or just after (during the 3 months post-treatment) radiotherapy due to tumor progression, despite treatment. Tumor progression during the treatment was responsible for a deterioration of the general state, requiring the stop of the treatments and leading the patient to need palliative care.

Forty-six (78%) patients completed CRT and the second diffusion-weighted MRI scan was performed at 3 months after completing CRT for all of them.

Patient clinical characteristics, tumor characteristics and therapeutic management are detailed in Table 1. Mean age was 60.8 ± 12.1 years old. Most patients were male (84.7%) and had active tobacco consumption (71.2%). All patients have unique location HNSCC.

Among patients with oropharyngeal cancer, ADC1 primary tumor value was significantly greater in patients with positive HPV status (HPV+) than in patients with negative HPV status (HPV-) (0.76 ± 0.17 versus 0.64 ± 0.13; *p* = 0.03).

Ten patients (17%) had benefits from 3 cures of induction chemotherapy, consisting of a basic regimen of cisplatin (75 mg/m^2^), docetaxel (75 mg/m^2^) and 5-fluorouracil (750 mg/m^2^), otherwise known as TPF (Taxane, Platine, 5-Fluoro-uracil), per course of treatment. 

All patients underwent on-tumor-site radiotherapy or radio-chemotherapy with a prescribed dose of 70 Gy (2 Gy per fraction) in Simultaneous Integrated Boost, with, in case of radio-sensitization, 3 courses of cisplatin 100 mg/m^2^ or Epithelial Growth Factor Receptor (EGFR)-targeted therapy, such as cetuximab, weekly (400 mg/m^2^ loading dose 8 days prior to radiotherapy, followed by a course of 250 mg/m^2^ weekly, concurrent with radiotherapy). Full-dose radiotherapy was administered to the tumor site in 51 patients (86%) and reduced-dose radiotherapy was administered to 8 patients (14%) because of toxicity grade 3 or 4 (6 patients (11%)) or deterioration of general condition (2 patients (3%)). Among the 8 patients who did not have access to full-dose therapy at the tumor site, 8 (100%) died during or just after (during the 3 months post-treatment) radiotherapy due to tumor progression, despite treatment.

The inter-observer reproducibility, as assessed by the ICC, was high, with ICC = 0.92 [95% CI: 0.87–0.94] for the first diffusion-weighted MRI (before RCT) and ICC = 0.98 [95% CI: 0.97–0.99] for the second diffusion-weighted MRI (after CRT completion).

### 3.2. Factors Associated with Tumor Residue

Among the 46 patients completing CRT, 19 (41.3%) had tumor residue. Twelve patients (43.5%) had macroscopic tumor residue on clinical examination. Twenty-five patients (54.3%) had tumor residue on the second diffusion-weighted MRI. Among the 25 patients with tumor residue on the second diffusion-weighted MRI, squamous cell carcinoma was confirmed histologically based on biopsy results in 19 patients (76%). 

Univariate and multivariate analysis of factors associated with post-CRT tumor residue are summarized in Table 2. In univariate analysis, only ADC1 primary tumor value was significantly associated with tumor residue (0.56 ± 0.11 in patients with tumor residue versus 0.79 ± 0.13 in patients without tumor residue; *p* < 0.0001) (Figure 3a). When ADC1 was dichotomized at the median, ADC1 lower than 0.7 was significantly more frequent in patients with tumor residue (73.7% versus 11.1%, *p* < 0.0001). In multivariate analysis, ADC1 less than 0.7 was the only factor significantly associated with tumor residue (OR = 22.6 [4.9–103.6], *p* < 0.0001). 

Among the 46 patients completing CRT, 25 patients had oropharyngeal cancer. Among them, 9 (36.0%) had tumor residue. Univariate and multivariate analyses of factors associated with post-CRT tumor residue among the 25 patients with oropharyngeal cancer were performed. In univariate analysis, ADC1 primary tumor value was significantly associated with tumor residue (0.59 ± 0.08 in patients with tumor residue versus 0.81 ± 0.09 in patients without tumor residue; *p* = 0.0009). When ADC1 was dichotomized at the median, ADC1 lower than 0.7 was significantly more frequent in patients with tumor residue (7/8 patients (87.5%) versus 2/17 patients (11.8%), *p* = 0.0005). HPV status was not significantly associated with tumor residue (2/12 patients with positive HPV status (16.7%) versus 7/13 patients with negative HPV status (53.8%); *p* = 0.10). Variables included in multivariate analysis were ADC1 less than 0.7 and HPV status. In multivariate analysis, ADC1 less than 0.7 was the only factor significantly associated with tumor residue (OR = 53.8 [4.8–600.3], *p* = 0.001).

### 3.3. Factors Associated with Post-Treatment Disease Progression or Early Recurrence

A total of 27 patients with no tumor residue remained part of the study follow-up for a 12 months period. Among them, 1 (3%) patient was lost to follow-up. Early recurrence or disease progression was observed in 6 (23%) of the 26 patients. At the end of study follow-up, 20 (74.1%) patients survived without recurrence. 

Univariate analysis of factors associated with post-treatment disease progression or early recurrence is summarized in Table 3. 

The only factor significantly associated with disease progression or early recurrence was the delta ADC (*p* = 0.0009) (Figure 3B). 

When ADC variation was dichotomized at the median, patients with ADC variation greater than 0.7 had time of disease-free survival significantly longer than patients with ADC variation lower than 0.7 (377.5 [286–402] days versus 253 [198–370], *p* < 0.0001) (Figure 4).

Among the 26 patients with no tumor residue and followed up for a 12-month period, 16 had oropharyngeal cancer. Early recurrence or disease progression was observed in 4 (25.0%) of the 16 patients. The only factor significantly associated with disease progression or early recurrence was the delta ADC (*p* = 0.0001). When ADC variation was dichotomized at the median, patients with ADC variation greater than 0.7 had time of disease-free survival significantly longer than patients with ADC variation lower than 0.7 (380 [286–402] days versus 253 [198–353], *p* = 0.02). Patients with positive HPV status tended to have longer time of disease-free survival than patients with negative HPV status (348 [253–396] days versus 289 [198–402] days; *p* = 0.05).

## 4. Discussion

The present study thus makes compelling arguments in support of the benefits of diffusion-weighted MRI as a powerful predictive tool for post-CRT early recurrence or tumor residue in HNSCC. 

Our study highlights the fact that an increase in ADC (delta ADC) between initial pre-treatment MRI (ADC1) and follow-up MRI at 3 months (ADC2) of less than 0.7 was associated with disease progression or recurrence, and therefore necessitates more intense monitoring. 

Furthermore, our study demonstrates that ADC1 values of less than 0.7 also predict post-CRT tumor residue, thus indicating treatment failure, contrary to ADC1 above this threshold that significantly predicts a positive response. This second point is indeed in accordance with the existing literature [37]. 

To our knowledge, this is the only prospective study that has assessed diffusion-weighted MRI for this indication and yielded highly significant results with ADC or delta ADC thresholds that clearly differentiate relapse-free survival probability. 

Our findings corroborate the conclusions of retrospective trials that have assessed diffusion-weighted MRI in the detection of residual disease or recurrence in head and neck cancers. Some authors [44] have reported diffusion coefficient sensitivity at 94.6% and specificity at 95.9% at 6 months post-CRT. However, the study in question involved interpretation of precancerous mucosal lesions, regional lymphatic nodes and low-risk mucosal irradiation. In the same vein, other authors have reported sensitivity at 84% and specificity at 90%, although there was strong variability in tumor histology and irradiation procedures [45]. Lastly, other authors have produced less impressive results, with sensitivity at 67% and specificity at 86% in irradiated laryngeal and hypopharyngeal cancer [46].

Two noteworthy prospective studies were conducted during the post-CRT monitoring period. A prospective trial involving 29 patients assessed the percentage variance of tumor diffusion coefficients pre-treatment and at 3 weeks post-treatment using two-year locoregional control as the primary endpoint. The authors reported sensitivity and specificity at 89% [47]. The second trial assessed variance of the same diffusion coefficients in the periods before, during and after treatment (at 6 months post-treatment) in patients suffering from head and neck cancers and treated with CRT or radiotherapy alone. Reduction in ADC during or after treatment was 100% specific for recurrence within 6 months. Post-treatment ADC was more predictive of 12 month regional control than pre-treatment ADC or ADC during treatment, with sensitivity at 80% and specificity at 100% [48].

On the histologic point, our findings can be interpreted with more necrosis and cell death during therapy. Some authors also showed that ADC reflects the underlying microenvironment of cancers [31]. The positive correlation of ADC and stromal component could suggest that the poor prognostic value of high pretreatment ADC might partly be attributed to the tumor-stroma component, a known predictor of local failure [31].

Our study also demonstrated a correlation between ADC and HPV status with a higher ADC within an HPV+ tumor. On the contrary, some authors showed that ADC mean and median were statistically significantly lower in p16-positive tumors [49]. Others did not find any correlation between ADC and HPV status [32].

It is also of interest to compare these results with those of an alternative imaging technique that is under assessment for the purposes of monitoring: positron emission tomography combined with computed tomography (PET/CT). Several studies have investigated post-treatment monitoring, based on the hypothesis that a significant decrease in mean SUV between pre- and post-therapeutic periods was suggestive of reduced post-treatment glucose metabolism in tumor cells. Certain authors have suggested that PET/CT is more sensitive than CT alone in detecting post-radiotherapy disease persistence [50]. Others have emphasized that all patients whose post-radiotherapy SUV was less than 3.0 were free from viable residual tumor cells [51]. It is, however, equally important to note that SUV measurement may be affected by variables such as imaging system and its calibration, dosage of the markers used, time elapsed since irradiation and patient serum glucose levels, such that SUV alone is inappropriate for assessing tumor response. Furthermore, radiation-induced changes may also result in increased SUV [52]. The purpose of a currently ongoing French therapeutic trial is to highlight the benefits of intensive monitoring, with emphasis on the inclusion of several PET/CT scans, by comparison with conventional monitoring methods. 

Moreover, diffusion-weighted MRI would appear to be more efficient than PET/CT in earlier and more specific detection of persistent diseases. Indeed, several studies have compared these two monitoring procedures [51] along with the advantages of combining the two [53,54], but not from a prospective point of view. 

Such early detection of recurrence or disease progression has a direct impact on patient management in terms of both overall survival and comorbidity. Indeed, retrospective data indicates that locoregional control is enhanced by faster salvage treatment [55,56,57]. Likewise, postoperative complication rates linked to surgical salvage would appear to rise in proportion to time elapsed due to an increase in radiation-induced fibrosis [55,57,58,59].

One of the limitations of this study was its small sample size with low event rate and absence of external validation. Nevertheless, the effects that were measured had sufficient power to yield functional, practical results that were able to be exploited statistically and corroborated the results of similar prior retrospective studies. 

A further limitation arises from the heterogeneity of the head and neck cancer sites. On the basis of the promising results yielded by the present study, it would be of interest to engage in larger-scale research to determine more homogenous populations in terms of cancer sites, given that each site has its own individual prognostic factors. 

## 5. Conclusions

The present study makes compelling arguments for the use of pre- and post-chemo-radiotherapy diffusion-weighted MRI in head and neck cancers. This technique brings about differentiation of patients with high potential for recurrence who necessitate intensive post-CRT monitoring. It also facilitates detection of early recurrence or disease progression, thus enabling faster, more efficient implementation of salvage treatment. Finally, diffusion-weighted MRI could be a useful tool to predict failure before CRT. Prospective studies with more inclusions would be necessary to validate our results.

## Figures and Tables

**Figure 1 cancers-12-01234-f001:**
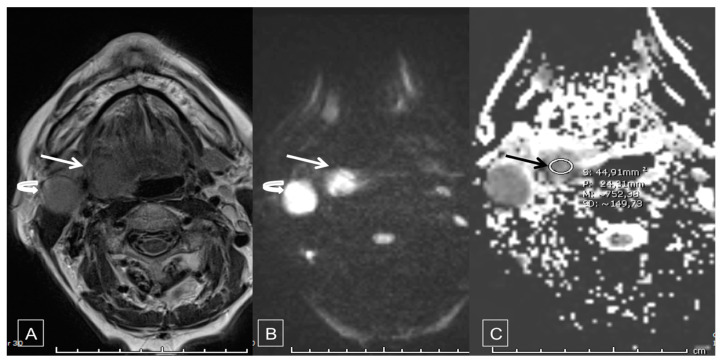
75-year-old female patient before treatment: (**A**) Axial T2, (**B**) high b-value (b = 1000), diffusion-weighted Magnetic Resonnance (MR) images show a 2 cm large lesion on the right tonsil extending anteriorly to the posterior border of the base of the tongue (arrow), along with a 2 cm large adenopathy in the II B area (curved arrow). These lesions exhibit markedly high signal intensity in image (**B**). (**C**) Apparent diffusion coefficient (ADC) map show restriction of diffusion in both the lymph node and the lesion, with a mean value of the ADC of 0.75 s/mm^2^.

**Figure 2 cancers-12-01234-f002:**
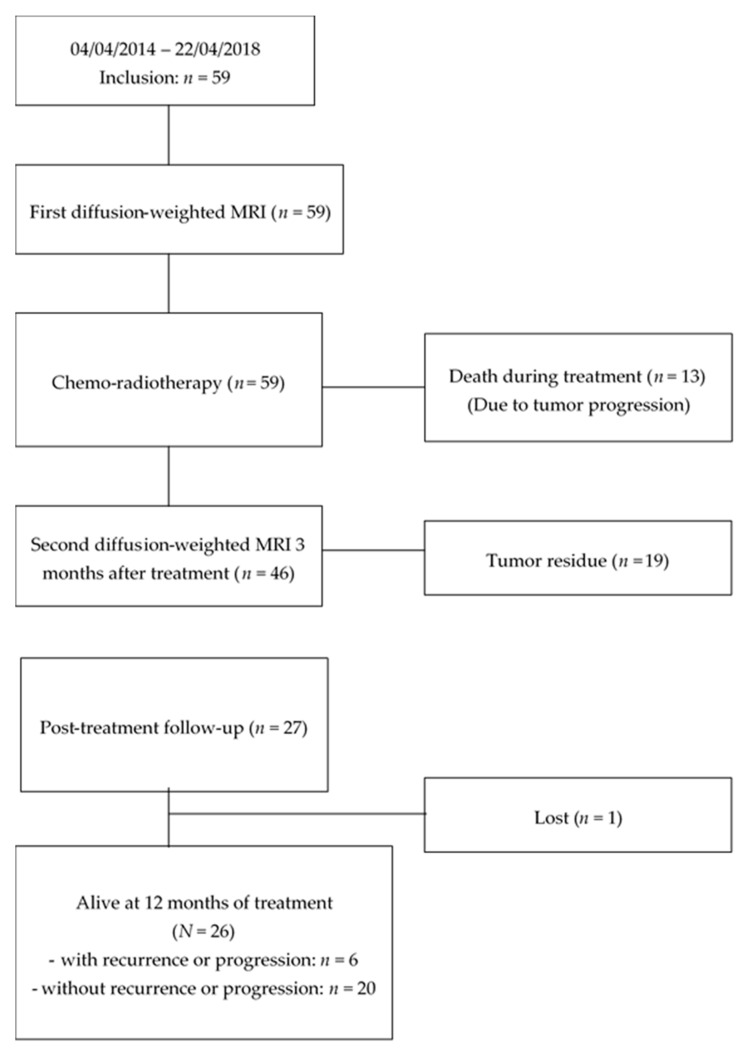
Flow-chart of the study.

**Figure 3 cancers-12-01234-f003:**
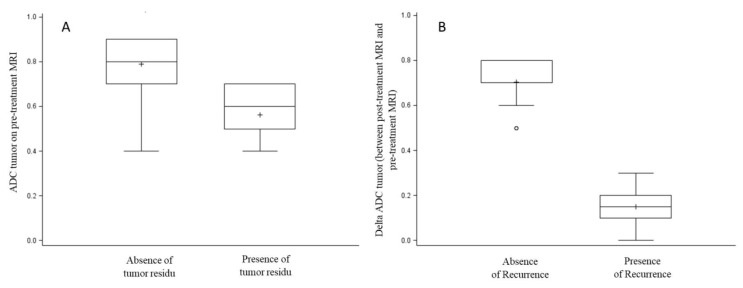
Factors associated with tumor residu (**A**) and early recurrence (**B**). (**A**) Box plot presenting ADC tumor value on pre-treatment MRI according to tumor residu, (**B**) Box plot presenting delta ADC tumor between post-treatment MRI and pre-treatment MRI according to recurrence. Each box shows median (as line), quartiles and mean (as cross). The whiskers represent values below the first quartile and above the third quartile within the 1.5-fold inter-quartile range respectively, and outliers beyond the whiskers are shown as circles.

**Figure 4 cancers-12-01234-f004:**
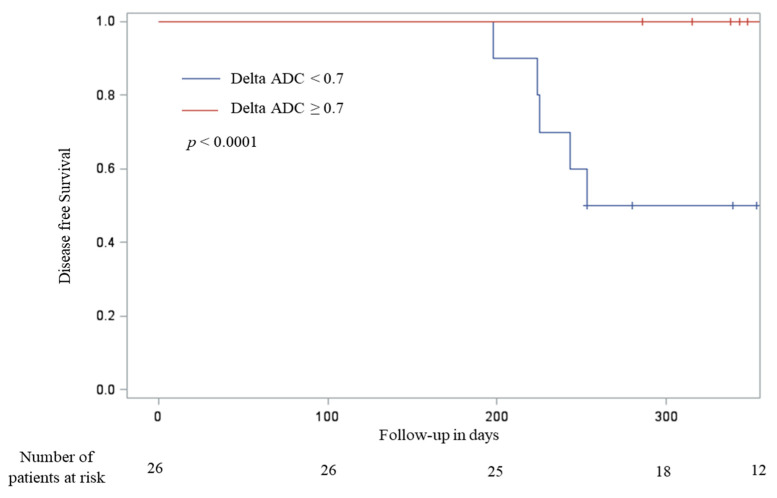
Recurrence-free survival according to tumor delta ADC (greater or lower than 0.7).

**Table 1 cancers-12-01234-t001:** Baseline clinical characteristics of patients, characteristics of tumors and treatment modalities decided.

Characteristics ^1^	Value
Patients	
Age at inclusion (year) − Mean ± SD	60.8 ± 12.1
Sexe Ratio (M/F)	50/9
Tobacco use	
No	8 (13.6)
Yes	51 (86.4)
Active	42 (82.4)
weaned	9 (17.6)
Alcohol	25 (42.4)
Body Mass Index (kg/m2) − Mean ± SD	23.9 ± 4.9
Head and Neck Tumor	
Unique location	59 (100)
Location	
Oropharynx	33 (55.9)
Oral cavity	8 (13.6)
Larynx	7 (11.9)
Hypopharynx	8 (13.6)
Cavum	1 (1.7)
Sinus	1 (1.7)
Parotid gland	1 (1.7)
HPV status (among oropharynx location)	
Positive	13/33 (39.4)
Negative	20/33 (60.6)
TNM classification	
T1	4 (6.8)
T2	5 (8.5)
T3	9 (15.2)
T4	41 (69.5)
N0	9 (15.2)
N1	8 (13.6)
N2a	1 (1.7)
N2b	10 (17.0)
N2c	22 (37.3)
N3	9 (15.2)
M0	57 (96.6)
M1	2 (3.4)
Histology	
Squamous cell carcinoma	59 (100)
Différentiation	
Well	28 (47.5)
Middle	24 (40.7)
Poor	7 (11.9)
Therapeutic sequence decided	
Induction chemotherapy	10 (17.0)
Radiotherapy	59 (100)
Exclusive	19
With Cetuximab	25
With Cisplatine	15

^1^ Data are presented as *n* (%) unless otherwise indicated.

**Table 2 cancers-12-01234-t002:** Univariate and multivariate analysis of factors associated with the presence of a tumor residue after chemo-radiotherapy.

Variables ^1^	Variables ^1^	Absence of Residue *N* = 27	Presence of Residue *N* = 19	Univariate Analysis	Multivariate Analysis ^2^	
				*p* Value	OR [CI 95%]	*p* Value
Tumor diameter on MRI (mm) − Mean ± SD		44.3 ± 18.8	52.6 ± 21.0	0.20		
Median ADC tumor (s/mm^2^) − Mean ± SD		0.79 ± 0.13	0.56 ± 0.11	<0.0001		
Patients with ADC tumor ≥0.7		24 (88.9)	5 (26.3)	<0.0001	22.6 [4.9–103.6]	<0.0001
<0.7		3 (11.1)	14 (73.7)			
Tumor location	Oral cavity	2 (7.4)	4 (21.0)	0.23		
	Oropharynx	16 (59.3)	9 (47.4)			
	Larynx	5 (18.5)	1 (5.3)			
	Hypopharynx	4 (14.8)	2 (10.5)			
	Sinus	0 (0.0)	1 (5.3)			
	Cavum	0 (0.0)	1 (5.3)			
	Parotid	0 (0.0)	1 (5.3)			
Initial T status	T1	2 (7.4)	1 (5.3)	0.64		
	T2	3 (11.1)	1 (5.3)			
	T3	6 (22.2)	2 (10.5)			
	T4	16 (59.3)	15 (79.0)			
Initial N status	N0	3 (11.1)	6 (31.6)	0.13		
	N+	24 (88.9)	13 (68.4)			
Initial M status	M0	27 (100.0)	18 (94.7)	0.41		
	M+	0 (0.0)	1 (5.3)			
Induction chemotherapy	Yes	5 (18.5)	3 (15.8)	0.99		
	No	22 (81.5)	16 (84.2)			
Therapeutic sequence	Radiotherapy exclusive	6 (22.2)	5 (26.3)	0.18		
	Radiotherapy with Cetuximab	10 (37.1)	11 (57.9)			
	Radiotherapy with Cisplatine	11 (40.7)	3 (15.8)			

^1^ Data are presented as *n* (%) unless otherwise indicated. ^2^ Factors included in the multivariate analysis were: ADC tumor ≥ 0.7, initial *N* status and therapeutic sequence.

**Table 3 cancers-12-01234-t003:** Factors associated with progression or early recurrence after chemo-radiotherapy in univariate analysis.

Variables	Variables ^1^	Time of Disease-Free Survival (Days) ^1^	Univariate Analysis *p* Value
Tumor diameter at first MRI (mm) Mean ± SD			0.31
Delta ADC (ADC2–ADC1) Mean ± SD			0.0009
Patients with Delta ADC (ADC2–ADC1)	≥0.7	377.5 [286–402]	<0.0001
	<0.7	253 [198–370]	
Induction chemotherapy	Yes	343 [253–396]	0.82
	No	353 [198–402]	
Therapeutic sequence	Radiotherapy exclusive	336 [243–386]	0.35
	Radiotherapy with Cetuximab	338 [198–396]	
	Radiotherapy with Cisplatin	370 [225–402]	
Tumor location	Oral cavity	342.5 [315–370]	0.71
	Oropharynx	346 [198–402]	
	Larynx	348 [243–391]	
	Hypopharynx	383 [343–394]	
Initial T status	T1	319.5 [253–386]	0.69
	T2	381 [280–389]	
	T3	340.5 [243–391]	
	T4	357 [198–402]	
Initial N status	N0	350 [198–402]	0.28
	N+	300 [243–357]	
Initial M status	M0	350.5 [198–402]	-
	M+	-	

^1^ Data are presented as median [range].

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
