# Peer review of "Predictive Value of Early Post-Treatment Diffusion-Weighted MRI for Recurrence or Tumor Progression of Head and Neck Squamous Cell Carcinoma Treated with Chemo-Radiotherapy"

_cancers, 2020, doi:10.3390/cancers12051234_

Round 1

Reviewer 1 Report

Thank you for the oportunity to review this interesting prospective study to determine the benefit of DWI to predict tumor progression in HNSCC.

The results are interesting and of potential interest to researchers in the field.

Yet, I have some concerns to adress.

The patient sample is relatively small, 13 patients died during the study time. Please specifiy the causes of these deaths.

Introduction:

Please change the first passage from the France point of view into a globally one and use English literature as a reference.

Please rewrite the sentence "The more a lesion is aggressive, the more it is cellular, and its ADC value is correspondingly lower: the lesion is visible on diffusion sequences as a clear hyper intensive signal. The apparent diffusion coefficient (ADC), which quantifies water molecule diffusion, can then be
 calculated"

There are no studies, which directly confirm your statement that there is a clear correlation between cellularity and aggressiveness.

Change diffusion sequences into high b-value images derived from DWI. 

You could add the reference by Surov et al. Oncotarget. 2017 May 10;8(35):59492-5949, which assessed the relationship between DWI and cellularity.

Materials and methods:

Please add a figure of a MRI of a patient.

Please provide a new figure 1 because it is very blurry.

You should perform subanalysis for the different treatment regimes. There are patients with an induction chemotherapy, some without, patients with only radiotherapy. This might confound your results.

Most of your patients had oropharyngeal cancer. Please provide the HPV status of these patients, as it is a very important prognostic factor. The ADC value are also correlated with HPV status (see for example de Perrot T et al. AJNR Am J Neuroradiol. 2017 Nov;38(11):2153-2160).

Regarding the DWI sequence, please provide more details for the analysis. Was only 1 region of interest drawn? How was it drawn? Did you exclude necrosis? In recent days, a whole lesion measurement is considered mandatory. Moreover, please provide the interreader variability by a second reader, as the ADC measurement can be difficult, especially after treatment.

Results:

A low pretreatment ADC value is already known to predict a worse outcome of the patients (for example Garbajs M et al. Radiol Oncol. 2019 Mar 3;53(1):39-48).

Please provide a box plot graph to visualize the comparison between patients with residual tumor and none in regard of delta ADC.

These findings are interesting and can be interpreted with more necrosis and cell death during therapy.

Discussion:

Well written.

You could add a short passage regarding correlations between histopathology and ADC values (see for example Driessen JP, et al. Radiology. 2014 Aug;272(2):456-63.; Meyer HJ et al. Magn Reson Imaging. 2018 Dec;54:214-217.; Surov A et al. Oncotarget. 2018 May 4;9(34):23599-23607)

Conclusion:

please delete "groundbreaking technique". This sounds quite a bit exaggerating.

Author Response

The authors would particularly like to thank the 2 reviewers who took time to read the article and to suggest corrections or details that allow a better understanding of our study and a better development of the results.

The authors thank the reviewers for their interest in this study and particularly appreciated the help they showed, in particular by proposing articles and expert literature on the subject which made it possible to specify the methodology better and to place better this article in the current scientific context.
Furthermore, it is very interesting to be able to have access to such a level of remarks when designing an article such as this one.

The authors have tried to respond as best they can, point by point, to the different questions of the reviewers and hope that the details will be sufficient to be accepted in your review.

Several manipulations, in particular the second reading of the MRI and HPV status of the patients were carried out within the recommended time.

Below is the point-by-point response to reviewer's comments by color code:
black: reviewers' comments

red: authors' responses by specifying the line number of the manuscript.

Please see the attachment in wich each modification was added to the text, yellow highlighted,

The entire article was proofread and corrected by an English native speaker.
All the statistics and methodology were produced under the direction of the clinical research unit and assistance with the statistical methodology of our center.
A random verification of 10% of the files was carried out during the prospective inclusion protocol.

The authors are at the disposal of the reviewers for any further clarifications they consider necessary.

Reviewer 1/

Thank you for the oportunity to review this interesting prospective study to determine the benefit of DWI to predict tumor progression in HNSCC.
The results are interesting and of potential interest to researchers in the field.
Yet, I have some concerns to adress.

The patient sample is relatively small, 13 patients died during the study time. Please specifiy the causes of these deaths.

The 13 patients died due to a tumor progression during treatment responsible for a deterioration of the general state requiring the stop of the treatments and the passage in palliative care. The reason for death was specified in the lines 161 and 162 manuscript and the flow chart (line 189)

Introduction:

Please change the first passage from the France point of view into a globally one and use English literature as a reference.

Epidemiological data has been updated with global and recent data based on recent English literature (2017/2020).Lines 38 and 39 : "Head and neck cancer is the fifth most common cancer, representing 5.3% of all cancers, with 890 000 new cases worldwide in 2017 (1,2)."

  1. Aupérin A. Epidemiology of head and neck cancers: an update. Curr Opin Oncol. 2020;32(3):178–86.
  2. Fitzmaurice C, Allen C, Barber RM, Barregard L, Bhutta ZA, Brenner H, et al. Global, regional, and national cancer incidence, mortality, years of life lost, years lived with disability, and disability-adjusted life-years for 32 cancer groups, 1990 to 2015: a systematic analysis for the global burden of disease study. JAMA Oncol. 2017;3(4):524–48.

You could add the reference by Surov et al. Oncotarget. 2017 May 10;8(35):59492-5949, which assessed the relationship between DWI and cellularity.

Thank you very much for this very clear and very interesting reference. We used the entire literature of this expert author on this topic to rewrite the introduction (lines 58 to 73). Several articles from this team were cited and this allowed us to argue our results also in the discussion.

Please rewrite the sentence "The more a lesion is aggressive, the more it is cellular, and its ADC value is correspondingly lower: the lesion is visible on diffusion sequences as a clear hyper intensive signal. The apparent diffusion coefficient (ADC), which quantifies water molecule diffusion, can then be calculated"

There are no studies, which directly confirm your statement that there is a clear correlation between cellularity and aggressiveness.

This sentence and the whole paragraph have been modified to appear more clearly in the article (lines 58 to 73). The modifications were carried out on the basis of the literature of the team of Surov A. et al.

Change diffusion sequences into high b-value images derived from DWI.

All the requested changes have been made in the article.

Materials and methods:

Please add a figure of a MRI of a patient.

Figure 1 has been added to the material and method (lines 116 to 124).

Please provide a new figure 1 because it is very blurry.

This figure became Figure 2 and the quality was increased because it was indeed very blurry (line 189).

You should perform subanalysis for the different treatment regimes. There are patients with an induction chemotherapy, some without, patients with only radiotherapy. This might confound your results.

As recommended, associations between treatment regimes (induction chemotherapy and therapeutic sequence) and tumor residue or post treatment disease progression or early recurrence have been studied (table 1 line 190).

Concerning tumor residue, neither induction chemotherapy nor therapeutic sequence were significantly associated with tumor residue. 3/19 patients (15.8%) with tumor residue had induction chemotherapy versus 5/27 patients (18.5%) without tumor residue had induction chemotherapy; p=0.99. Among patients with tumor residue: 5/19 (26.3%) had radiotherapy exclusive, 11/19 (57.9%) had radiotherapy and cetuximab, 3/19 (15.8%) had radiotherapy and cisplatine versus among patients without tumor residue : 6/27 (22.2%) had radiotherapy exclusive, 10/27 (37.1%) had radiotherapy and cetuximab, 11/27 (40.7%) had radiotherapy and cisplatine ; p=0.18. These results are now presented in table 2 (line 221).

Concerning post-treatment disease progression or early recurrence, neither induction chemotherapy nor therapeutic sequence were significantly associated post-treatment disease progression or early recurrence. Median time survival was 343 [253 – 396] days in patients with induction chemotherapy versus 353 [198 – 402] days in patients without induction chemotherapy; p=0.82. Median time survival was 336 [243 – 386] days in patients with radiotherapy exclusive, 338 [198 – 396] days in patients with radiotherapy and cetuximab, 370 [225 – 402] in patients with radiotherapy and cisplatine; p=035. These results are now presented in table 3 (line 235).

Most of your patients had oropharyngeal cancer. Please provide the HPV status of these patients, as it is a very important prognostic factor. The ADC value are also correlated with HPV status (see for example de Perrot T et al. AJNR Am J Neuroradiol. 2017 Nov;38(11):2153-2160).

HPV status of all oropharyngeal cancers was sought and HPV status of patients with oropharyngeal cancer is now presented in table 1 (line 190). 13/33 patients (39.4%) had positive HPV status.

Among patients with oropharynx tumor, HPV status and ADC1 primary tumor value were significantly associated : ADC1 primary tumor value was significantly greater in patients with positive HPV status than in patients with negative HPV status (0.76 ± 0.17 versus 0.64 ± 0.13; p=0.03). This result is now presented in the results paragraph (lines 169 to 171 ; table 1 line 190 ; lines 209 to 219 ; lines 255 and 262).

Univariate analysis of factors associated with early recurrence or disease progression among the 16 patients with oropharyngeal cancer, no tumor residue and follow-up for 12 months has been performed. Delta ADC greater than 0.7 was significantly associated with early recurrence or disease progression (p=0.02). HPV status tended to be associated with early recurrence or disease progression (p=0.05). No multivariate analysis has been performed because of the too small patients number.

Regarding the DWI sequence, please provide more details for the analysis. Was only 1 region of interest drawn? How was it drawn? Did you exclude necrosis? In recent days, a whole lesion measurement is considered mandatory.

Thank you for this request for clarification.
The ADC coefficient was calculated by two different radiologists with experience of more than 5 years in imaging the head and neck. The readings were performed blindly so as not to influence the result.
The measurement of the ADC was carried out on the whole of the tumor including on the areas of necrosis.
The measurement was not made on a volume but on the surface of the tumor on 3 sections with the largest diameter in axial sections.
3 measurements were performed for each tumor and the ADCmean coefficient was chosen (lines 128 to 132 ; 138 to 139 ; 185 to 187)

Moreover, please provide the interreader variability by a second reader, as the ADC measurement can be difficult, especially after treatment.

Interreader concordance has been studied using intra class correlation coefficient. Two analyses have been realized: one concerning ADC evaluation on MRI before treatment and one concerning ADC evaluation on MRI after treatment. In the 2 cases, interreader concordance was good: ICC = 0.92 [95% CI: 0.87-0.94] for the first diffusion-weighted MRI (before RCT) and ICC=0.98 [95% CI: 0.97-0.99] for the second diffusion-weighted MRI (after CRT completion). Lines 136 and 137 ; Lines 185 to 187.

Results:

A low pretreatment ADC value is already known to predict a worse outcome of the patients (for example Garbajs M et al. Radiol Oncol. 2019 Mar 3;53(1):39-48).

Thank you for this reference which was added in discussion of the article. Lines 273 to 276.

Please provide a box plot graph to visualize the comparison between patients with residual tumor and none in regard of delta ADC.

As suggested, a new figure 3 showing ADC tumor on the initial MRI according to presence or absence of tumoral residue (figure 3A) and delta ADC tumor between post-treatment MRI and pre-treatment MRI according to presence or absence of recurrence (figure 3B) has been added. Lines 203 to 205 ; 241 to 250.

These findings are interesting and can be interpreted with more necrosis and cell death during therapy.

Discussion:

Well written.

You could add a short passage regarding correlations between histopathology and ADC values (see for example Driessen JP, et al. Radiology. 2014 Aug;272(2):456-63.; Meyer HJ et al. Magn Reson Imaging. 2018 Dec;54:214-217.; Surov A et al. Oncotarget. 2018 May 4;9(34):23599-23607)

Thank you for these references which allowed the addition of a discussion paragraph on the correlation between the histology and the ADC coefficient. Lines 298 to 306.

Conclusion:

please delete "groundbreaking technique". This sounds quite a bit exaggerating.

"groudbreaking technique" was removed from conclusion as requested. line 344

Reviewer 2/

I thank both authors for this manuscript about ADC and recurrence in H&N cancers treated by exclusive chemo-radiotherapy.
I want to discuss some minors and majors issues

Introduction:

Authors should better described results from other type of cancers and prediction by ADC value.

A new paragraph has been written, presenting the predictive contribution of the ADC coefficient in the treatment of ovarian, rectal, breast and brain tumor. Lines 63 to 73

Methods:

HPV status should be recorded as it is one of the best known pronostic factors for H&N cancers and could be associated with ADC value.

HPV status of all oropharyngeal cancers was sought and HPV status of patients with oropharyngeal cancer is now presented in table 1 (line 190). 13/33 patients (39.4%) had positive HPV status.

Among patients with oropharynx tumor, HPV status and ADC1 primary tumor value were significantly associated : ADC1 primary tumor value was significantly greater in patients with positive HPV status than in patients with negative HPV status (0.76 ± 0.17 versus 0.64 ± 0.13; p=0.03). This result is now presented in the results paragraph (lines 169 to 171 ; table 1 line 190 ; lines 209 to 219 ; lines 255 and 262).

Univariate analysis of factors associated with early recurrence or disease progression among the 16 patients with oropharyngeal cancer, no tumor residue and follow-up for 12 months has been performed. Delta ADC greater than 0.7 was significantly associated with early recurrence or disease progression (p=0.02). HPV status tended to be associated with early recurrence or disease progression (p=0.05). No multivariate analysis has been performed because of the too small patients number.

Is ADC acquired in one slide, 3d-ROI or all the volume? If not acquired on the overall volume, how authors deal with reproducibility of delineation?

Thank you for this request for clarification.
The ADC coefficient was calculated by two different radiologists with experience of more than 5 years in imaging the head and neck. The readings were performed blindly so as not to influence the result.
The measurement of the ADC was carried out on the whole of the tumor including on the areas of necrosis.
The measurement was not made on a volume but on the surface of the tumor on 3 sections with the largest diameter in axial sections.
3 measurements were performed for each tumor and the ADCmean coefficient was chosen (lines 128 to 132 ; 138 to 139 ; 185 to 187)

Interreader concordance has been studied using intra class correlation coefficient. Two analyses have been realized: one concerning ADC evaluation on MRI before treatment and one concerning ADC evaluation on MRI after treatment. In the 2 cases, interreader concordance was good: ICC = 0.92 [95% CI: 0.87-0.94] for the first diffusion-weighted MRI (before RCT) and ICC=0.98 [95% CI: 0.97-0.99] for the second diffusion-weighted MRI (after CRT completion). Lines 136 and 137 ; Lines 185 to 187.

It is not cleared why patients have only 12 months follow-up but survival analysis show results for more than 12 months (moreover when you see that survival outcomes are differents only after nearly 12 months). Time from the beginning of calculation of EFS should be described. I'm also very surprised about the lack of recurrence or progression during nearly the first year.

Thanks for this comment. Patients have been followed up 12 months after completing CRT. The Kaplan Meyer curve was wrong because of time of follow up had been calculated from date of diagnosis. This point has been corrected and the time of follow up is now calculated from the date of completing RCT. A corrected Kaplan Meyer curves according delta ADC (greater or lower than 0.7) is now showed in Figure 4. Lines 258 to 265.

Results:

Results for survival analysis are not depending time? (0% versus 80%? Is that a Khi 2?) For log rank, survival results are usually given depending time (1y OS, 2years OS ….). VPP, VPN SE and SP are usually not used for prognostic factor of outcome but only for diagnosis because they don't take into account time depending variable and time depending status. Moreover population seems to be to

Results of survival analysis are now presented using time of disease free survival in a new Table 3. Line 235 and 255 to 262

VPP, VPN, Se and Sp of ADC variation greater than 0.7 have been deleted of results paragraph.

Title and figures:

EFS is used as a title for one axis of the Survival graph but should be replaced by progression free Survival or disease free Survival as EFS means both progression/relapse/symptoms G> 3/death free Survival according NCI definition, moreover number of patient at risk should be described in the text or graph as it seems that only 9 patients presented delta ADC < 0.7 and ones could argue that no statistical analysis should be done in case of less than 10 patients in one group Bollschweiler, 2003; Pocock et al., 2002).
The axis title has been modified (“Disease free survival”. The number of patients at risk has been added in the figure 4. Line 264

As suggested and because survival curves should be interpreted with caution if fewer than 10 patients leave (Bollschweiler 2003, Brookmeyer 1982), follow up for Kaplan Meyer curve has been stopped at 360 days (time at which 12 patients were still at risk).

Reviewer 2 Report

I thank both authors for this manuscript about ADC and recurrence in H&N cancers treated by exclusive chemo-radiotherapy.

I want to discuss some minors and majors issues

Introduction:

Authors should better described results from other type of cancers and prediction by ADC value

Methods:

HPV status should be recorded as it is one of the best known pronostic factors for H&N cancers and could be associated with ADC value.

Is ADC acquired in one slide, 3d-ROI or all the volume ? If not acquired on the overall volume, how authors deal with reproducibility of delineation ?

It is not cleared why patients have only 12 months follow-up but survival analysis show results for more than 12 months (moreover when you see that survival outcomes are differents only after nearly 12 months). Time from the beginning of calculation of  EFS should be described. I'm also very surprised about the lack of recurrence or progression during nearly the first year.

Results:

Results for survival analysis are not depending time ? (0% versus 80% ? Is that a Khi 2 ?) For log rank, survival results are usually given depending time (1y OS, 2years OS ….). VPP, VPN SE and SP are usually not used for pronostic factor of outcome but only for diagnosis because they don't take into account time depending variable and time depending status. Moreover population seems to be to

Title and figures:

EFS is used as a title for one axis of the Survival graph but should be remplaced by progression free Survival or disease free Survival as EFS means both progression/relapse/symptoms G> 3/death free Survival according NCI definition, moreover number of patient at risk should be described in the text or graph as it seems that only 9 patients presented delta ADC < 0.7 and ones could argue that no statistical analysis should be done in case of less than 10 patients in one group Bollschweiler, 2003; Pocock et al., 2002).

I think that authors should better clarified methods and also work with a dedicated statistical unit.

Author Response

(The authors gave the same response as above.)

Round 2

Reviewer 1 Report

Thank you for the extensive editing of the manuscript. I think it has improved significantly and is ready for publication!

Author Response

Reviewer 1/

Thank you for the extensive editing of the manuscript. I think it has improved significantly and is ready for publication!

Dear reviewer,
Thank to you for your extremely relevant comments and your bibliographic help which significantly improved the quality of the article. We are honored that it is now up to your requirements for publication.

Reviewer 2/

I'm still concern about the number of patients but the manuscript is now Ok for publication.

Conclusion in abstract should be edit (still "groundbreaking" word in it) and authors should to be more carefull about results from such a short sample without validation

Conclusion of the manuscript and abstract should include that these results need to be check in external cohort.

Dear reviewer,

Thank you for this second review and your methodological contribution which enabled better development of the results.
As requested, the conclusion of the abstract and of the manuscript has been modified to be more moderate and cautious, in particular with the need for external validation of our results.

Lines 31 to 33 ; 337 and 338 ; 354 and 355

Reviewer 2 Report

I thank authors for Editing.

I'm still concern about the number of patients but the manuscript is now Ok for publication.

Conclusion in abstract should be edit (still "groundbreaking" word in it) and authors should to be more carefull about results from such a short sample without validation

Conclusion of the manuscript and abstract should include that these results need to be check in external cohort.

Author Response

(The authors gave the same response as above.)
